# The Assessment of the Association of Proton Pump Inhibitor Usage with Chronic Kidney Disease Progression through a Process Mining Approach

**DOI:** 10.3390/biomedicines12061362

**Published:** 2024-06-19

**Authors:** Kaile Chen, Farhad Abtahi, Hong Xu, Carlos Fernandez-Llatas, Juan-Jesus Carrero, Fernando Seoane

**Affiliations:** 1Department of Clinical Science, Intervention and Technology, Karolinska Institutet, 17177 Stockholm, Sweden; farhad.abtahi@ki.se (F.A.); cfllatas@itaca.upv.es (C.F.-L.); fernando.seoane@ki.se (F.S.); 2Department of Biomedical Engineering and Health Systems, School of Engineering Sciences in Chemistry, Biotechnology and Health, KTH Royal Institute of Technology, 14157 Huddinge, Sweden; 3Department of Clinical Physiology, Karolinska University Hospital, 17176 Stockholm, Sweden; 4Division of Clinical Geriatrics, Department of Neurobiology, Care Sciences and Society (NVS), Karolinska Institutet, 17177 Stockholm, Sweden; hong.xu.2@ki.se; 5Institute of Information and Communication Technologies (SABIEN-ITACA), Universitat Politècnica de València, Camino de Vera S/N, 46022 Valencia, Spain; 6Department of Medical Epidemiology and Biostatistics, Karolinska Institutet, 17177 Stockholm, Sweden; juan.jesus.carrero@ki.se; 7Department of Medical Technology, Karolinska University Hospital, 17176 Stockholm, Sweden; 8Department of Textile Technology, University of Borås, 50190 Borås, Sweden

**Keywords:** eGFR trajectory, process mining, multistate model, proton pump inhibitors (PPIs), H2 blockers (H2Bs), chronic kidney disease (CKD), longitudinal data analysis

## Abstract

Previous studies have suggested an association between Proton Pump Inhibitors (PPIs) and the progression of chronic kidney disease (CKD). This study aims to assess the association between PPI use and CKD progression by analysing estimated glomerular filtration rate (eGFR) trajectories using a process mining approach. We conducted a retrospective cohort study from 1 January 2006 to 31 December 2011, utilising data from the Stockholm Creatinine Measurements (SCREAM). New users of PPIs and H2 blockers (H2Bs) with CKD (eGFR < 60) were identified using a new-user and active-comparator design. Process mining discovery is a technique that discovers patterns and sequences in events over time, making it suitable for studying longitudinal eGFR trajectories. We used this technique to construct eGFR trajectory models for both PPI and H2B users. Our analysis indicated that PPI users exhibited more complex and rapidly declining eGFR trajectories compared to H2B users, with a 75% increased risk (adjusted hazard ratio [HR] 1.75, 95% confidence interval [CI] 1.49 to 2.06) of transitioning from moderate eGFR stage (G3) to more severe stages (G4 or G5). These findings suggest that PPI use is associated with an increased risk of CKD progression, demonstrating the utility of process mining for longitudinal analysis in epidemiology, leading to an improved understanding of disease progression.

## 1. Introduction

Chronic kidney disease (CKD) poses a significant global health challenge, affecting millions worldwide as it gradually impairs kidney function [1]. The estimated global prevalence of chronic kidney disease (CKD) was approximately 9% in 2017, representing a 29.3% increase since 1990 [2]. CKD also became the 12th leading cause of death in 2017, rising from 17th place in 1990 [2]. Glomerular filtration rate decline is a surrogate endpoint in evaluating the progression of kidney disease [3]. Empirical evidence underscores the important role of eGFR point estimate as a robust prognosticator for adverse CKD events, notably end-stage renal disease (ESRD) [4]. Longitudinal analysis of eGFR trajectories and retrospective analysis of eGFR slopes are also important for understanding the dynamics of CKD progression and predicting patient outcomes [4,5,6]. Some studies have relied on the absolute change in eGFR as a surrogate [7,8] for kidney function progression, but they have failed to capture the rich insights that can be derived from analysing multiple eGFR measurements. 

Proton pump inhibitors (PPIs) are widely prescribed for acid-related gastrointestinal disorders [9]. PPI usage greatly reduces stomach acid levels for longer periods [10,11,12] compared to H2 blockers (H2Bs). However, many observational investigations have indicated PPI usage with kidney adverse events, such as an increased risk of CKD, CKD progression and acute kidney injury (AKI) [8,13,14]. While most observational studies have shown associations between PPIs and CKD, some studies have reported no significant association [15] or only a weak link [16]. The underlying mechanism of PPI usage with adverse effects on CKD remains the subject of an ongoing inquiry. Plausible explanations include the induction of acute interstitial nephritis (AIN) [17,18] and hypomagnesemia [19] by the usage of PPIs. Delayed treatment or incomplete recovery from AIN can lead to acute kidney disease, potentially increasing the risk of developing CKD. H2B [20] is another class of medication used to treat gastrointestinal conditions but has rarely been reported to be associated with nephrotoxicity.

Process discovery, a process mining technique for discovering and analysing sequences of events over time, has gained traction in healthcare for studying clinical pathways and disease trajectories [21]. For instance, one study successfully predicted in-hospital mortality using a process mining-based methodology [22]. Additionally, researchers in [23] modelled treatment pathways for patients with cancer using process mining techniques. In a different application, process mining facilitated the development of dynamic risk models for chronic conditions [24], resulting in simplified process maps that enhance the understanding and measurement of disease progression. Furthermore, the study [25] employed process mining to uncover disease trajectories in periodontitis patients, enabling researchers to monitor the progression of related multi-morbidities. However, its application in analysing the progression of chronic diseases within epidemiological studies remains scarce. Pharmacoepidemiology research often applies cohort designs and drug new-user designs to assess the association between exposures and outcomes [26]. These designs take the importance of time sequence and derive more robust evidence for causal inference compared to cross-sectional or case-control studies. Process mining excels at visualising such time-ordered events, making it a valuable tool for exploring disease trajectories. This capability enables us to use process mining to extract longitudinal eGFR data and compare the effects of PPIs and H2Bs on the risk of CKD progression.

This study contributes to the fields of nephrology and epidemiology by applying a novel approach based on process mining. Firstly, it employs a process mining technique for the discovery, visualisation and comprehension of CKD patients’ trajectories over time. Additionally, the study seeks to provide evidence regarding the association between the use of PPIs and the accelerated progression of CKD through an extensive longitudinal analysis.

In this retrospective cohort study, we aim to evaluate the usefulness of the process mining approach in epidemiological research by extracting eGFR-mediated longitudinal trajectories to assess the association between exposure to PPIs and CKD progression in comparison to exposure to H2Bs.

## 2. Materials and Methods

Section 2.1 details the study design and resources utilised in this research. Following this, Section 2.2 outlines the population inclusion criteria adopted for the study. Section 2.3 describes the primary methodology of process mining discovery. Subsequently, Section 2.4 provides a comprehensive overview of the statistical methods employed in this study.

### 2.1. Study Design and Resources

This study employed a retrospective drug new-user cohort design utilising data from the Stockholm Creatinine Measurements (SCREAM) cohort, a population-based cohort in Sweden [27]. The study period was from 1 January 2006 to 31 December 2011. A drug new-user cohort was constructed, comprising PPI new users as the analysis group and H2B new users as the active-comparator group. The use of PPIs and H2Bs was assumed to be continuous from initiation to the study endpoints. Identification of PPIs, H2Bs and other concomitant medications was accomplished through Anatomical Therapeutic Classification (ATC) codes (Appendix A). Comorbidities were identified using ICD-10 codes (Appendix A). This study was carried out following the rules of the Declaration of Helsinki. The Swedish National Board of Welfare and the Stockholm Regional Ethics Review Board approved this study (Ethical Approval Code 2017/793-31).

### 2.2. Study Population

Based on the study aims and design, the study population was selected on five specified inclusion and exclusion criteria. (1) We included PPI or H2B new users. The new user was defined as the same as a prior classic epidemiology study by Klatte [8] (i.e., an individual who has received their first PPI or H2B dispensation between 1 January 2007 and 31 December 2010 was considered a new user; the data in 2006 were used to ascertain no prescription of the medication before, and the data in 2011 were used to make sure at least 1 year follow-up). (2) At least 1 time serum creatinine (Scr) test less than 6 months before the index date. (3) New users of PPIs or H2Bs with chronic kidney disease (CKD), defined by a baseline eGFR less than 60 mL/min/1.73 m^2^. The baseline eGFR value was the latest measurement before or on the date of medication prescription. (4) People ≥ 18 years old. Exclusion: Individuals with a baseline eGFR of less than 15 mL/min/1.73 m^2^ were excluded, as they were in the end stage of CKD.

The eGFR was calculated based on the Chronic Kidney Disease Epidemiology Collaboration (CKD-EPI) 2009 creatinine equation [28] in adults aged 18 and older, which uses serum creatinine, age, ethnicity and gender. Although ethnicity data were unavailable due to legal restrictions, the misclassification of eGFR is expected to be minimal because most residents in the Stockholm region are of Caucasian origin. All creatinine measurements were conducted using either the enzymatic method or the corrected Jaffe method (alkaline picrate reaction) across three laboratory companies in the Stockholm region: Aleris, Unilabs and Karolinska [27]. Both inter- and intra-laboratory variations are considered minimal, as these laboratories undergo frequent quality audits and harmonisation checks by the national organisation EQUALIS (www.equalis.se accessed on 7 May 2024) [29].

### 2.3. Process Mining Discovery

The core methodology was based on the interactive process mining process [30]. The process mining discovery technique from the bupaR package [31] in R was used to derive process models and indicators [32]. Event logs were created for both PPI and H2B users. 

The event was built on the eGFR stages, comprising five stages that align with KDIGO categories [33,34]: G1 (normal or high, eGFR ≥ 90 mL/min/1.73 m^2^), G2 (mildly decreased, eGFR 60–89 mL/min/1.73 m^2^), G3A(mildly to moderately decreased, eGFR 45-59 mL/min/1.73 m^2^), G3B (moderately to severely decreased, eGFR 30–44 mL/min/1.73 m^2^), CKD4 (severely decreased, eGFR 15–29 mL/min/1.73 m^2^) and CKD5 (kidney failure, eGFR < 15 mL/min/1.73 m^2^). The follow-up period started from the date of PPI or H2B dispensation until either emigration, death or the conclusion of the study on 31 December 2011.

### 2.4. Statistical Analysis

Baseline demographic and clinical characteristics of the patients were summarised using frequencies and percentages for categorical variables, means with standard deviations (SDs) for normally distributed continuous variables, and medians with interquartile ranges (IQR) for continuous variables that are not normally distributed.

The process indicator described the eGFR trajectories, including the frequency and transition time. Transition time is the duration between two events, also referred to as sojourn time in this analysis. It can be understood as the interval between two measurements or the observed time it takes for eGFR to shift from one stage to another.

Age, gender, comorbidities and concomitant medications were recorded on the index date. Comorbidities included gastrointestinal diseases, cardiovascular and cerebrovascular diseases, diabetes mellitus and chronic obstructive pulmonary disease, with detailed ICD-10 codes provided in Appendix A. Concomitant medications, such as NSAIDs, statins and antithrombotics, have detailed ATC codes available in Appendix A.

We employed a multistate model [35] to analyse the transition probabilities between different eGFR stages identified through the process indicator. This model incorporated PPI/H2B use, age, gender, comorbidities and concomitant medications as covariates to account for their potential influence on the transitions between eGFR stages. After adjusting for these covariates, the multistate model produced adjusted hazard ratios with corresponding 95% confidence intervals (CIs) to estimate the effect of PPI/H2B use on eGFR decline. The specific model structure incorporating these multiple covariates is presented in Figure 1. All statistical analyses and data visualisations were conducted using R 4.3.1.

## 3. Results

A total of 12,043 newly initiated users of PPIs and H2Bs with CKD were included in this investigation, comprising 11,486 PPI users and 557 H2B users (Figure 2). The baseline characteristics showed a median age of 81 years for the PPI cohort and 78 years for the H2B cohort (Table 1), indicating more elderly individuals in the former. Moreover, both PPI and H2B cohorts exhibited a higher proportion of females than males (57% and 65%, respectively). Regarding baseline eGFR, the PPI cohort displayed a lower median compared to the H2B cohort. Additionally, the PPI group has a greater prevalence of comorbidities compared to the H2B group.

### 3.1. Process Indicator

The process indicators (Figure 3 and Figure 4), which describe the transition of eGFR stages of the PPI and H2B groups over the follow-up period, were structured. These indicators are sequentially ordered from the node “start” to “end”; directional arrows denote the progression from antecedent to consequent events, with nodes representing eGFR stages. Node shading correlates with event frequency, displaying both relative and absolute frequencies. Additionally, arrow annotations depict relative frequencies and median transition times. 

Figure 3 and Figure 4 illustrate the transition process among five eGFR stages for PPI and H2B groups, respectively. Both groups initially presented at stage 3 (G3) or stage 4 (G4), aligning with our inclusion criteria of CKD patients (eGFR < 60 and ≥15 mL/min/1.73 m^2^). Three arrows emanating from the “start” node represent the baseline eGFR distribution within each group. For instance, in the PPI group, 58.2% started at G3A, 30.5% at G3B and 11.3% at G4. The H2B group had a higher proportion starting at G3A (68.6%) and lower proportions starting at G3B (25%) and G4 (6.5%) compared to the PPI group. The stage connected to the “end” represents the last observed event. Transitions among eGFR stages can be visually compared between the two groups in Figure 3 and Figure 4. Notably, the PPI group exhibited a higher proportion of patients progressing to more severe stages (G4 or G5, e.g., G3A → G4 is 7% in the PPI group and 5% in the H2B group) and fewer instances of improvement, suggesting a potentially negative impact of PPI usage on CKD progression. Conversely, individuals in the H2B group relatively experienced more favourable transitions, with a higher proportion transitioning back to better eGFR stages (indicating improved kidney function such as G1 and G2, e.g., G3A → G2 is 41% in the PPI group and 47% in H2B group) compared to the PPI group. 

Due to the unstructured nature and complexity of Figure 3 and Figure 4 and to facilitate the comparison of eGFR trajectories between PPI and H2B users, eGFR stages were integrated to simplify the process indicators, as illustrated in Figure 5. Specifically, G1 and G2 were merged into a singular event denoted as “G1 or G2”, G3A and G3B were combined into “G3”, and G4 and G5 were consolidated into “G4 or G5”.

Figure 5 illustrates the distribution of transitions between eGFR stages between PPI and H2B groups that exhibited notable disparities. Notably, 33% of PPI users experienced a reduced eGFR from “G3” to “G4 or G5”, compared to 25% of H2B users who underwent this transition within the same median duration. After declining to “G4 or G5”, relatively more H2B users transitioned back to “G3” than PPI users. Similarly, 2.19% of PPI users worsened from “G1 or G2” to “G4 or G5”, while only 1.8% of H2B users experienced this decline. Furthermore, in the improvement transition from “G3” to “G1 or G2”, 45% of PPI users underwent this change, whereas 51% of H2B users experienced the transition. Moreover, in the H2B group, a larger percentage of individuals remained at “G1 or G2” (28%) and “G3” (60.3%) compared to PPI users, where the corresponding figures were slightly lower, at 23.6% and 55.7%, respectively. Conversely, in the PPI group, a higher proportion of individuals stayed at “G4 or G5” (20.7%) compared to the H2B group (11.7%).

### 3.2. The Patterns of eGFR Trajectories

Each trajectory signifies the changes in individual eGFR stages throughout follow-up. The length of each trajectory corresponds to the number of eGFR stages observed during follow-up. Notably, individuals utilising PPI exhibit longer eGFR trajectory lengths compared to those in the H2B group. Additionally, PPI users have a shorter median follow-up duration in contrast to H2B users (Table 2).

Compared to the H2B group, the PPI group shows a 75% higher hazard (HR: 1.75, 95% CI: 1.49, 1.52) for the progression from eGFR stage “G3” to stages “G4 or G5” (G3 → G4/5, as shown in Table 2). There was no statistically significant distinction in the transition from “G3” to “G1/2”, “G1/2” to “G3” and “G4/5” to “G3” between patients being administered PPIs versus those receiving H2Bs. 

Subgroup analysis to compare the effects of PPIs versus H2Bs on eGFR trajectories across different subgroups is presented in Appendix A.

### 3.3. Integrating Process Mining with Epidemiological Design versus Classic Epidemiology Methods

A detailed comparison with the study by Klatte et al. is presented in Table 3. Our study’s main finding aligns with Klatte’s research, providing evidence that the process mining approach can yield results comparable to those of classical epidemiological methods. We applied process mining techniques to achieve a more comprehensive understanding of disease progression. Unlike the previous study that focused on the association between exposure (PPI) and outcome (eGFR decline of 30%), we constructed eGFR trajectories by capturing all observed eGFR stages. This approach allows for a different perspective analysis by comparing the properties of these trajectories between individuals exposed to PPIs and those exposed to H2Bs.

## 4. Discussion

We applied a process mining approach to extract the eGFR trajectory patterns, followed by comparing the patterns between PPI and H2B groups. The results showed that the pattern of eGFR trajectory is more complex in PPI users than in H2B users. In addition, the PPI group has a significantly higher risk of undergoing from eGFR stage G3 to the eGFR stage G4 or G5, representing a faster decline in kidney function than the H2B group. 

### 4.1. Process Mining in Chronic Disease Research

Process mining techniques have been applied in healthcare since 2001 [35], providing three main techniques: process discovery, conformance checking and enhancement [36]. These techniques are predominantly utilised to analyse data from electronic health records pertaining to clinical or treatment pathways to identify bottlenecks within hospitals and enhance overall efficiency [37]. While some previous studies have attempted to use process mining to trace disease trajectories in diabetes, sepsis and general diseases [38,39,40,41], our study initiated the attempt to synthesise process mining with epidemiological cohort design to assess the association between PPI usage and kidney function decline by comparing the eGFR trajectory patterns among CKD patients. 

### 4.2. Investigating the Association between PPI Use and CKD Progression through eGFR Trajectories

We used eGFR stages as the measurement events to investigate the patterns of eGFR in PPI users compared to H2B users. Specifically, the eGFR stages transit to higher eGFR stages signify the progression of CKD. Previous research primarily defines CKD progression by quantifying absolute eGFR changes, operating under the assumption of its linear progression [5]. However, the decline in kidney function in the population might have different patterns due to various factors that do not follow a steady linear decline. Some studies used the mixed model of linear quadratic, cubic and probabilistic clustering to produce trajectories to cope with the property of non-linearity [6] or the Bayesian smoothing method to define trajectories [42]. Applying process mining, in this case, provides dynamic changes with multiple measurements in eGFR over time since we observed the manifest difference in eGFR trajectories, which also represent the complexity of an individual. Notably, individuals in the PPI group exhibited more frequent serum creatinine measurements within a considerably shorter follow-up duration, indicative of heightened healthcare utilisation and worse baseline conditions, which could be a higher burden of comorbidities and older age, as corroborated by baseline characteristics. In the Stockholm region, PPI is generally the first-line treatment for gastrointestinal conditions like gastroesophageal reflux disease and ulcer disease [43]. 

Following adjustment for potential confounders such as baseline conditions of comorbidities, concomitant medications, baseline kidney function (baseline eGFR), sex and age, PPI users demonstrated a significantly higher rate of transition from G3 to G4 or G5 stages compared to H2B users, suggesting that PPI usage is associated with increased risk of CKD progression. These findings align with previous research indicating an association between PPI use and an increased risk of CKD progression [8,14,18]. Furthermore, we construed transitions to lower eGFR stages (i.e., larger eGFR values) as improvements or fluctuations, notwithstanding prevailing beliefs regarding the irreversibility of kidney function decline [44]. Contrary to the conventional notion of steady GFR progression, many CKD patients exhibit non-linear GFR trajectories or protracted periods of non-progression [42]. Our findings align with existing evidence regarding the non-linear eGFR changes, revealing transitions to improved kidney function (e.g., G3 to G1/G2 or G4/G5 to G3). Notably, a higher proportion of the H2B group experienced such improvements compared to the PPI group, although this difference did not reach statistical significance. Similarly, intermittent transitions from G1 or G2 to severe G3 showed no significant disparity between PPI and H2B cohorts. These transitions may signify eGFR fluctuations from G3 to G1/2, subsequently reverting to G3 or higher stages G4/G5. Other studies [5,45] have also posited that short-term increases in eGFR may stem from muscle mass loss or recovery from acute kidney injury.

### 4.3. The Process Mining Approach in This Study and the Comparison with Classic Epidemiological Research and Artificial Intelligence 

A comparison was conducted between the current study and the prior investigation to ascertain the utility and reproducibility of the process mining-assisted approach. This study also employed the cohort design [46], drug new-user design and active-comparator design [47] as Klatte’s study [8] in order to reduce selection bias. This study goes beyond replicating previous findings concerning the association between PPI usage and the increased risk of CKD progression. Combining process mining with these epidemiological designs introduces a new approach to epidemiological studies. This integration allows for a more comprehensive analysis and understanding of the data. To the best of our knowledge, this is the first application of interactive process mining to analyse longitudinal renal function trajectories in real-world epidemiological data. 

Process indicators were derived from the serum creatine test records and the timing of these tests, resulting in divergent patient patterns, including testing frequency variations indicative of baseline conditions. Consequently, these process indicators depicted the disease trajectories of patients from a distant point, with greater complexity in the process map potentially signalling severe conditions in reality. Compared to the property of classic epidemiology studies, which directly test the hypothesis of exposure and outcome, incorporating process mining into epidemiological settings may be less efficient than classical methods. However, process mining can provide valuable insights into the data landscape and illuminate associations from a broader perspective (in a process indicator/map). This underscores the usefulness of integrating process mining into epidemiological studies. 

We additionally constructed a multistate model based on the process indicator. The process indicator reflects real-world eGFR trajectories, offering a descriptive patient trajectory. Therefore, the initial relationship represented on process indicators might potentially be biased by confounding factors. Thus, multistate modelling allows for comparing the transition intensities between PPI and H2B groups with the adjustment of potential confounders. In healthcare research utilising process mining, it is common practice to integrate other statistical methods to test whether differences between two groups are attributable to factors beyond potential confounding variables or random errors [38]. 

Process mining, when combined with multistate models, presents a valuable tool in healthcare for patient journey modelling, disease progression tracking, and risk stratification. This hybrid approach, incorporating rule-based logic and statistical modelling elements, offers a high degree of interpretability, allowing for clear insights into patient transitions between distinct health states over time. However, unlike machine learning and deep learning methodologies, which excel at handling complex, high-dimensional data and uncovering intricate patterns, multistate models are inherently limited to scenarios with well-defined states and transitions. While deep learning techniques can process extensive amounts of unstructured data to predict outcomes, they often do so at the expense of transparency and interpretability. Consequently, multistate models are particularly suited for applications where understanding the underlying processes and maintaining model transparency are critical. In scenarios where complex data and predictive accuracy are paramount, machine learning and deep learning can be combined with multistate models to leverage the strengths of both approaches. For instance, machine learning can be used to enhance the predictive power and adaptability of multistate models.

### 4.4. Limitations, Challenges and Future Work

A limitation of this study is the high median age of the population. This is because the SCREAM dataset can only capture individuals who have accessed healthcare services, and younger individuals, who are generally healthier, are less likely to utilise healthcare resources. Consequently, our findings are representative of the real world, which includes people who take PPIs/H2Bs with CKD and have accessed healthcare utilisation within the Stockholm region. Additionally, analysing complex medical histories with multiple events challenges deriving meaningful associations. Domain knowledge is crucial to simplify this process and extract valuable insights.

Future work should focus on standardising the application of process mining in epidemiological research. There is potential for using process mining to investigate disease interactions, such as mediation and collider effects. Additionally, expanding the application of process mining to other diseases presents a promising avenue for future research. Fully leveraging process mining’s strengths in visualisation can automatically generate associations between diseases, diagnoses and medications, enabling comprehensive data-driven hypotheses.

## 5. Conclusions

This study evaluated the feasibility of applying process mining techniques to analyse and provide useful insights into real-world data. Process mining discovered the disease progression transitions in real-world data, which were used to construct a multistate model. By quantifying transition intensity using the multistate model, we identified a significantly higher risk of CKD progression in the PPI group compared to the H2B group. This is supported by a greater likelihood of eGFR trajectories advancing to higher stages in the PPI group. These findings confirm the prior evidence that PPI usage increases the risk for CKD progression and indicate that the process mining approach is beneficial for analysing longitudinal epidemiological data. 

## Figures and Tables

**Figure 1 biomedicines-12-01362-f001:**
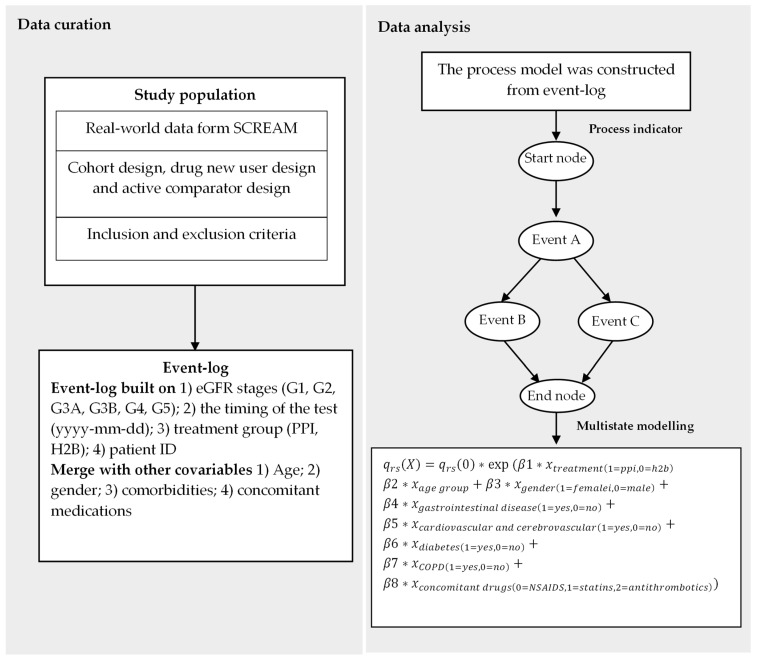
Diagram illustrating data curation and analysis in this study; data curation includes study design used for collection and evet-log build; data analysis describes the process mining approach and the algorithm utilised in the multistate model. In the algorithm, $q_{rs}\left(X\right)$ represents the transition intensity or the instantaneous rate from event r to event s. The term q_rs_(0) denotes the baseline transition intensity, while exp (β1) corresponds to the hazard ratio.

**Figure 2 biomedicines-12-01362-f002:**
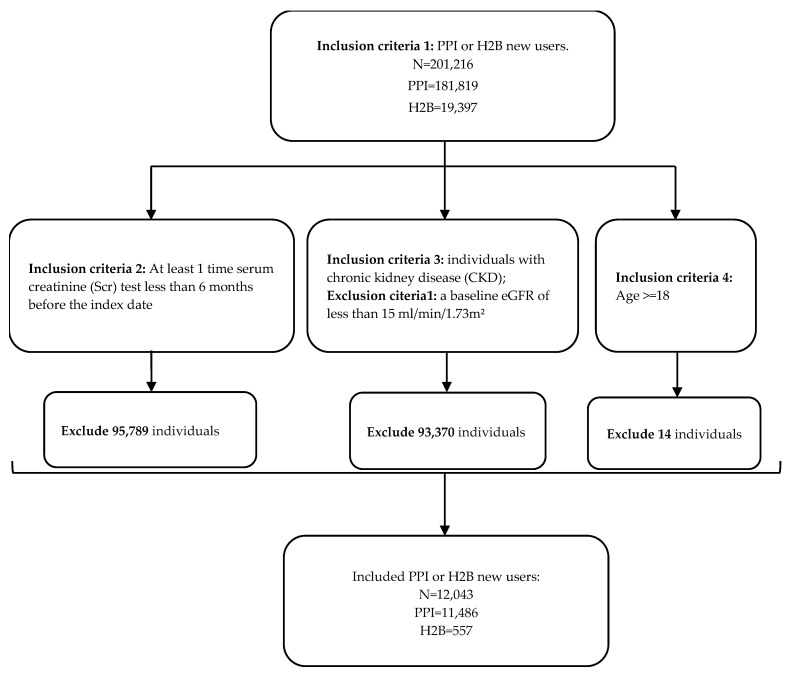
Flowchart of the selection process of study participants among new users of PPI and H2B, in accordance with the study’s inclusion and exclusion criteria.

**Figure 3 biomedicines-12-01362-f003:**
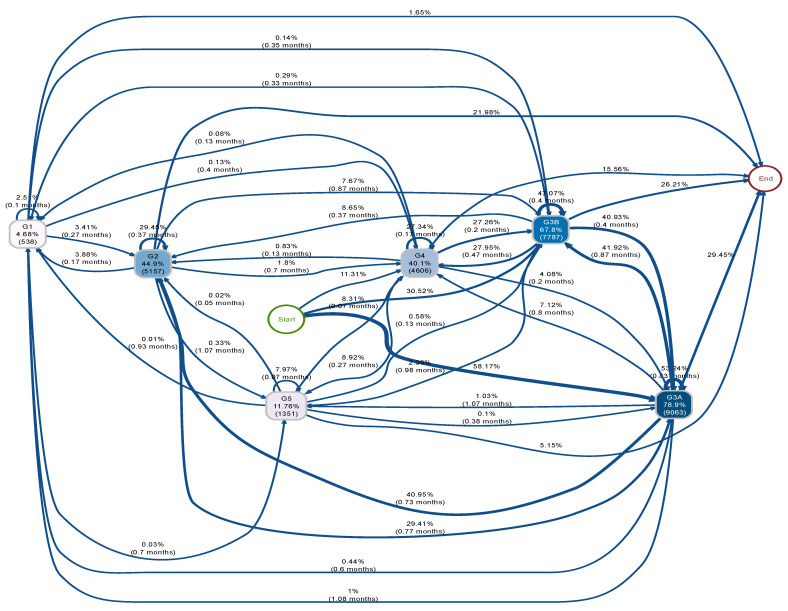
Initial process indicator of eGFR stage transitions over time in the PPI group. This figure illustrates the transition of eGFR stages in the PPI group over the follow-up period. Nodes represent event relative and absolute frequencies, with darker shades indicating higher event frequencies. Transition relative frequencies and median times are annotated on the arrows. Arrows signify time-ordered sequences of events, with varying thicknesses reflecting the frequency of transitions.

**Figure 4 biomedicines-12-01362-f004:**
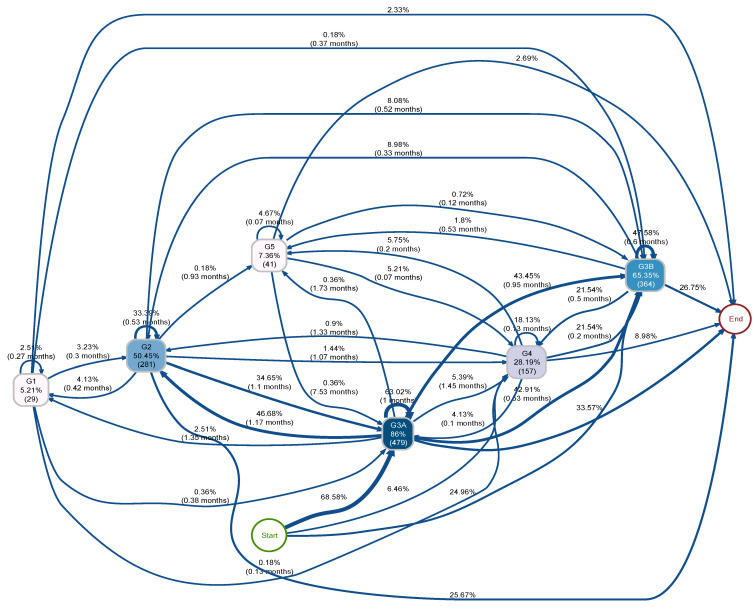
Initial process indicator of eGFR stage transitions over time in the H2B group. This figure illustrates the transition of eGFR stages in the H2B group over the follow-up period. Nodes represent event relative and absolute frequencies, with darker shades indicating higher event frequencies. Transition relative frequencies and median times are annotated on the arrows. Arrows signify time-ordered sequences of events, with varying thicknesses reflecting the frequency of transitions.

**Figure 5 biomedicines-12-01362-f005:**
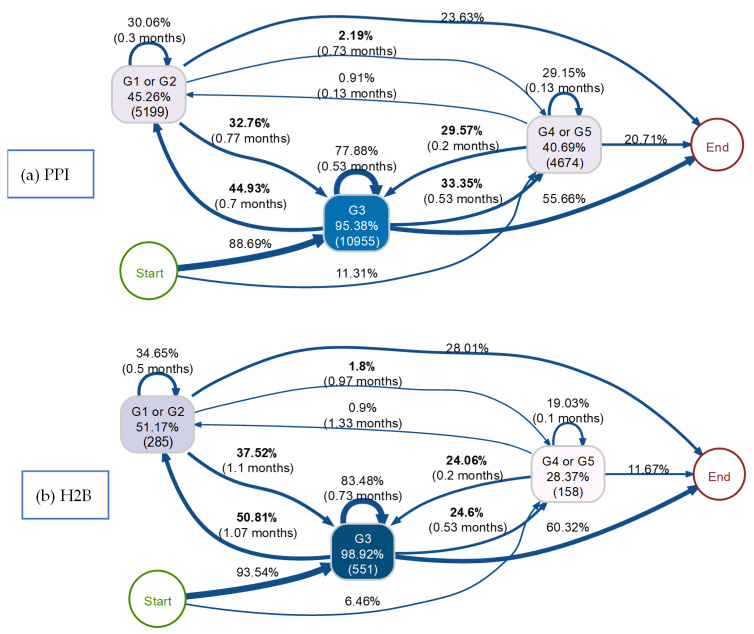
Process indicators of integrated eGFR stage transition over time between the PPI and H2B groups. (**a**) represents the PPI group; (**b**) represents the H2B group. Nodes display event relative and absolute frequencies, with darker shades indicating higher event frequencies. Transition relative frequencies and median times are depicted on the arrows. Arrows signify time-ordered sequences of events, with varying thicknesses reflecting the frequency of transitions. Transitions with proportions showing potential significant differences between the PPI and H2B groups are highlighted in bold.

**Table 1 biomedicines-12-01362-t001:** Characteristics of the baseline between new users of PPIs and H2Bs with CKD in Stockholm during 2006–2011.

	PPI N = 11,486	H2B N = 557
**Age, median (IQR)**	81 (73, 86)	78 (70, 85)
Age group, n%		
18~45	160 (1.4)	10 (1.8)
46~64	1109 (9.7)	74 (13.3)
65~80	4262 (37.1)	251 (45.1)
≥81	5954 (51.8)	222 (39.9)
Gender		
Female	6603 (57)	361 (65)
Male	4883 (43)	196 (35)
Comorbidities, n%		
Gastrointestinal diseases		
Gastroesophageal reflux disease	547 (4.8)	19 (3.4)
Upper gastrointestinal tract bleeding	488 (4.2)	10 (1.8)
Ulcer disease	1097 (9.6)	18 (3.2)
*H. Pylori* infection		
Cardiovascular and cerebrovascular diseases		
Myocardial infarction	2164 (18.8)	95 (17.1)
Cerebrovascular disease	2527 (22)	104 (18.7)
Peripheral vascular disease	1290 (11.2)	58 (10.4)
Congestive heart failure	1976 (17.2)	80 (14.4)
Hypertension	10,005 (87.1)	469 (84.2)
Diabetes mellitus	2743 (23.9)	133 (23.9)
Chronic obstructive pulmonary disease (COPD)	2042 (17.8)	94 (16.9)
Concomitant medication, n%		
NSAIDs, aspirin	4941 (43)	257 (46.1)
Statins	4179 (36.4)	217 (39)
Antithrombotics	7553 (65.8)	321 (57.6)
Baseline eGFR, median (IQR)	47.8 (38.2, 54.6)	50.7 (42.4, 55.9)
G3A (45–59 mL/min/1.73 m^2^)	6681(58.2)	382 (68.6)
G3B (30–44 mL/min/1.73 m^2^)	3506 (30.5)	139 (24.9)
G4 (15–29 mL/min/1.73 m^2^)	1299 (11.3)	36 (6.5)

**Table 2 biomedicines-12-01362-t002:** The comparison of eGFR trajectory patterns between PPI and H2B groups.

	PPI	H2B
Trajectory length, median (IQR)	11 (5, 21)	10 (5, 21)
Follow-up time(month), median (IQR)	23.3 (11.1, 37.7)	33.3 (17.7, 46.3)
eGFR stages transition:		
G3 → G1/2		
Number, %	5161 (44.9)	283 (50.8)
Median time (IQR), months	0.70 (0.13, 2.97)	1.07 (0.13, 4.08)
^a^ Hazard ratio, 95% CI	0.95 (0.78, 1.16)	Reference
G1/2 → G3		
Number, %	3763 (32.76)	209 (37.52)
Median time (IQR), months	0.77 (0.17, 2.70)	1.10 (0.20, 2.80)
^a^ Hazard ratio, 95% CI	0.92 (0.75, 1.12)	Reference
G3 → G4/5		
Number, %	3831 (33.35)	137 (24.60)
Median time (IQR), months	0.53 (0.13, 2.0)	0.53 (0.10, 2.13)
^a^ Hazard ratio, 95% CI	1.75 (1.49, 2.06)	Reference
G4/5 → G3		
Number, %	3396 (29.57)	134 (24.06)
Median time (IQR), months	0.20 (0.07, 0.83)	0.20 (0.07, 0.89)
^a^ Hazard ratio, 95% CI	1.13 (0.97, 1.31)	Reference

^a^. The hazard ratio and 95% CI were derived from multistate modelling after adjusting age, sex, comorbidities, concomitant medication and baseline eGFR.

**Table 3 biomedicines-12-01362-t003:** Summary of current study vs. a prior study that employed classic epidemiological methods.

	Klatte et al. [8]	Present Study
Study type	Classic epidemiology study	Process mining integrated with epidemiology design
Study design	Cohort; active-comparator design; new-user design	Cohort; active-comparator design; new-user design
Data source	SCREAM	SCREAM
Outcome	A doubling of serum creatinine or >30% decline in eGFR	Multiple eGFR stages
Measure	eGFR absolute changes; once a 30% decline or more in eGFR is observed, the follow-up will be discontinued	eGFR stages; collecting all observed eGFR stages until the end of the study
Analysis methods	Descriptive statistics; Cox proportional hazard regression model	Descriptive statistics; process indicators; multistate model
Study period	2006–2011	2006–2011
Study population	PPI or H2B new users with baseline eGFR ≥ 15 mL/min/1.73 m^2^	PPI or H2B new users with CKD (15 mL/min/1.73 m^2^ ≤ eGFR < 60 mL/min/1.73 m^2^)
Key results	PPI users had an increased risk for eGFR decline of 30% or more compared with H2B users	eGFR trajectory distribution differently between PPI users and H2B users; PPI users had an increased risk from eGFR ‘G3’ transit to ‘G4 or G5’

## Data Availability

The SCREAM contains sensitive personal data that cannot be publicly shared due to GDPR regulations. We welcome collaboration project proposals that adhere to GDPR and national and institutional regulations concerning data sharing and access. For inquiries, please contact juan.jesus.carrero.@ki.se.

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
