# Peer review of "The Assessment of the Association of Proton Pump Inhibitor Usage with Chronic Kidney Disease Progression through a Process Mining Approach"

_biomedicines, 2024, doi:10.3390/biomedicines12061362_

Round 1
Reviewer 1 Report
Comments and Suggestions for Authors
The paper entitled: “The assessment of the association of proton pump inhibitors use with chronic kidney disease progression through a process mining approach” by Chen K. et al deals with an innovative approach to epidemiological pharmacology studies that employs a “mining discovery technique” based on a bioinformatic algorithm in order to demonstrate that proton pump inhibitors (PPIs) protracted use can lead to chronic kidney disease progression.
This clinical hypothesis is actually debated and authors support it by analyzing data from an analysis of retrospective data.
The paper is therefore interesting in a twofold manner: first for the innovative bioinformatic approach that is allegedly superior to classical epidemiological approaches, and second for the clinical pharmacology inference they reach through the novel approach.
However, in our opinion, the paper has some weaknesses that must be addressed before considering its publication in a journal such as Biomedicines.
-
The abstract is both redundant and confounding. We suggest to better clarify at a glance the “process mining technique” in general, and in the specific case, in order to provide the readers with a better overview
-
The study compares the kidney toxicity effects of PPIs with that of H2B drugs. Are there other possible reference drugs to compare? Were potential other pharmacological interactions possible?
-
eGFR measurements were performed in the past over a long period of time. Were these measurements homogeneous? A detailed description of methods used, especially in case of method changes, should be helpful to validate the homogeneity of data.
-
line 121 “skewed data” we suggest to better explain which kind of skewness was present
-
line 122 “time spent on each transit” is a technical expression which should be better explained here
-
The information approach used in methods is not clearly understandable to us. A diagram presentation of the general and the specific approach could help.
-
The population median age was very high, even if comparable in the two groups. Discussion should address this limitation and a possible subgroup with a more significant median age should be possibly singled out.
-
As the authors point out the PPI group suffered from many more comorbidities than the other group. We are concerned that this observation is actually quite limiting the validity of the conclusions reached.
-
words in Figures 2,3 and 3 are too small and blurred to be well seen in a publication. Please make them bigger
-
Since the approach used probably falls into an algorithm to be used for artificial intelligence approaches to medicine in the future, could the authors elaborate on that?
Author Response
Hi,
Thank you very much for taking the time to review this manuscript. Please see the attached response file.

Reviewer 2 Report
Comments and Suggestions for Authors
A process mining approach was proposed to assess the association of proton pump inhibitor use with chronic kidney disease progression. I agree that research is needed on this topic; however, the paper requires a major revision to clarify and elaborate on some parts. Please refer to my comments as follows.
Comment 1. Update “Type of the Paper (Article)” to “Article”.
Comment 2. Abstract: Elaborate on the research results and impacts.
Comment 3. Keywords: Add more terms to better reflect the scope of the paper.
Comment 4. Section 1 Introduction:
(a) The authors should discuss the prevalence and severity of CKD.
(b) In the first three paragraphs, 27 references were cited, which are not appropriate. Please ensure sufficient explanation and elaboration are shared.
(c) The literature review is considered incomplete. Please provide a concise summary of the existing works' methodologies, results, and limitations. Mainly, focus on recent 5-year journal articles.
Comment 5. Section 2 Materials and Methods:
(a) Add an introductory paragraph before Subsection 2.1 to enhance the organization of the paper.
(b) Did the authors download the data from an open data repository?
(c) The innovation is weak if the authors followed the traditional process of the mining process.
(d) Incomplete presentation of the design and formulations of the methodology. Please provide equations, pseudo-codes, workflows, etc.
Comment 6. Section 3 Results:
(a) Figure 1 should be elaborated, particularly in the inclusion and exclusion criteria.
(b) Table 1, what are “(73,86)”, “70,85”, “(38.2,54.6)”, and “(42.4,55.9)”. In addition, carefully check the spacing, such as “6681(58.2)” and “382(68.6)”. Apply the same comment in Table 2.
(c) Enhance the resolutions of all figures. Please enlarge the file to 200% to confirm no content is blurred.
(d) Figures 2 to 4 are difficult to understand. Please ensure sufficient explanation is provided in the main text.
(e) The results and analysis showed basic statistics. More analysis is expected.
Comment 7. Performance comparison with the existing works is required.
Comment 8. Elaborate on future research directions.
There are some grammatical mistakes. In addition, please correct the issue of proper spacing between words and symbols.
Author Response
Hi,
Thank you very much for taking the time to review this manuscript.
Please see the attachment.

Round 2
Reviewer 1 Report
Comments and Suggestions for Authors
Author's have addressed point by point our previous criticisms. We think the paper has been changed considerably. In its present form the paper can indeed be considered for publication
Author Response
Hi,
Thank you very much for taking the time to review this manuscript. Your feedback is greatly appreciated.
Reviewer 2 Report
Comments and Suggestions for Authors
Although the quality of the paper is enhanced, there are some comments remain unaddressed:
Comment 1. Affiliations:
(a) There is a formatting issue in the first two lines.
(b) The emails of co-authors are missing.
Comment 2. The format in the list of references is not correct. Please check with the journal’s template.
Comment 3. The authors should clarify the research contributions by mentioning “research contributions” in the introduction.
Comment 4. The resolution of Figure 1 (workflow on the R.H.S.) should be enhanced.
Comment 5. Figure 2: The lowest arrow is not properly linking with the blocks.
Comment 6. More written descriptions are needed to better explain Figures 3-5.
Author Response

(The authors gave the same response as above.)
